# Kernel Matrix Estimation of a Determinantal Point Process from a Finite Set of Samples: Properties and Algorithms

**Marc Castella** *marc.castella@telecom-sudparis.eu*
*SAMOVAR, Télécom SudParis, Institut Polytechnique de Paris, 91120 Palaiseau, France*

**Jean-Christophe Pesquet** *jean-christophe.pesquet@centralesupelec.fr*
*OPIS Inria, CentraleSupelec, Université Paris-Saclay, France*

**Reviewed on OpenReview:** *https://openreview.net/forum?id=Cyx9LwB5IN*

## Abstract

Determinantal point processes (DPPs) on finite sets have recently gained popularity because of their ability to promote diversity among selected elements in a given subset. The probability distribution of a DPP is defined by the determinant of a positive semi-definite, real-valued matrix. When estimating the DPP parameter matrix, it is often more convenient to express the maximum likelihood criterion using the framework of $L$-ensembles. However, the resulting optimization problem is non-convex and $\mathcal{NP}$-hard to solve.

In this paper, we establish conditions under which the maximum likelihood criterion has a well-defined optimum for a given finite set of samples. We demonstrate that regularization is generally beneficial for ensuring a proper solution. To solve the resulting optimization problem, we propose a proximal algorithm which minimizes a penalized criterion. Through simulations, we compare our algorithm with previously proposed approaches, illustrating their differing behaviors and providing empirical support for our theoretical findings.

## 1 Introduction

Many modern application domains require to generate or promote diversity in samples. This is for example the case in document summarization (Lin & Bilmes, 2012), recommender systems (Zhou et al., 2010) but also in optimization and learning (Zhang et al., 2017). Spatial statistics is yet another domain where diversity can play an important role, for example for sensor placement (Krause et al., 2008), cellular networks (Miyoshi & Shirai, 2014; Torrisi & Leonardi, 2014). Determinantal Point Processes (DPPs) are precisely probabilistic models for random sets that can promote diversity. Indeed, DPPs are known to encode repulsion between elements, meaning that sets containing points that are close or similar to each other are less likely to occur.

DPPs already appeared in the 60's and were later identified in Macchi (1975) where they were called "fermion processes". They also naturally appear in several situations such as random matrix theory (Soshnikov, 2000) or quantum theory. Since then, they have been quite extensively studied by mathematicians (Hough et al., 2009; Borodin, 2015) and they appear in many places in probability, algebra, and combinatorics. Besides they have been considered both in a continuous setting as point processes, or in a discrete setting as subsets of a given finite set.

Recently, they have gained a renewed attention in the data science community (Kulesza & Taskar, 2012) and related applications such as recommendation systems (Gartrell et al., 2017), document or data summarization (Burt et al., 2020; Tremblay et al., 2019), image processing (Launay et al., 2021), random design (Dereziński et al., 2022),... A particularly appealing property of DPPs that can explain their increasing popularity is that many inference tasks can be performed efficiently. As a consequence, the problem of learning the parameters of a DPP appears as crucial although it is considered a challenging problem. A common approach relies

upon the maximum likelihood estimator. It has been conjectured by Kulesza in his thesis that the problem is $\mathcal{NP}$-complete, a proof of which recently appeared in Grigorescu et al. (2022). The explored solutions to this issue either try to learn the full underlying kernel or a few parameters encoding it.

The first class of approaches for inference on DPPs consider a parametrized form of the kernel and make specific assumptions (Kulesza & Taskar, 2011; Affandi et al., 2014; Bardenet & Titsias, 2015). To tackle the continuous case, such a parametrized approach has been the most natural; see e.g. Biscio & Lavancier (2017) for a minimum contrast estimation technique and other works from the same authors. In the second class of approaches, which aim to learn the full kernel, carefully designed methods are required. Gillenwater et al. (2014) proposed a first approach and in Affandi et al. (2014) a Bayesian MCMC method was used. Alternatively, methods based on optimization are likely to play a prominent role as described in the next section. A somewhat intermediate route consists in imposing a modeling structure on the kernel matrix. For example, a common framework assumes low rank matrices (Dupuy & Bach, 2016; Gartrell et al., 2017), but other structures exist (Mariet & Sra, 2016). Note that more general models have drawn attention and learning based on maximum likelihood has also been addressed for non symmetrical DPPs in Gartrell et al. (2019).

This paper addresses the problem of unparametrized DPP parameter estimation on finite sets, starting from a maximum likelihood (ML) formulation to learn a DPP kernel from an optimization perspective. We begin by showing that a minimizer of the ML objective does not necessarily exist. More precisely, the criterion infimum can only be approached when the norm of the estimated parameter goes to infinity. Beyond computational complexity, this issue suggests that alternative models may be better suited, or that one must settle for an approximate model. We here consider the latter possibility and explore the introduction of a regularization term. Regularization techniques are well-established in statistics and closely related to the Maximum a Posteriori (MAP) estimation framework (see, e.g., Bishop (2006); Hastie et al. (2009)). Within this context, we demonstrate the relevance of using proximal algorithms, which are well-suited for solving large-scale, potentially nonsmooth optimization problems where the objective splits into the sum of multiple terms. In view of the variety of penalization terms that can be handled by proximal methods, this paves the way to the development of more accurate estimation criteria.

## 1.1 Related works

The work in this paper has been motivated by works from different streams: optimization, DPP kernel parameter estimation, and DPP models in a lesser way.

Several optimization methods related to our problem have been developed, starting from the EM like method (Gillenwater et al., 2014) which tries to infer the marginal kernel of the DPP. Adopting the slightly less general model of an $L$-ensemble, the (log)-likelihood criterion simplifies. A particularly efficient algorithm based on a fixed-point iteration has been proposed in Mariet & Sra (2015) and interpreted as a minimization-maximization like algorithm. In Kawashima & Hino (2023), based on the latter viewpoint, a tighter local majorizer has been derived and an accelerated version of the algorithm has been introduced. However, difficulties in the convergence of the iterates were not deeply investigated. Both works only deal with full-rank DPPs, whereas some other works try to promote some structure (Mariet & Sra, 2016). The low-rank assumption remains the most common assumption explicitly made (Gartrell et al., 2017; 2018). Yet another trend in this line of works is to introduce some regularization terms in the criterion (Dupuy & Bach, 2016). The additional penalization term has been reported to allow good predictive performance of the estimated model in Gartrell et al. (2017). For continuous DPPs, a nonparametric estimation based on kernel methods has also been proposed and has led to similar criteria in Fanuel & Bardenet (2021). The mentioned works are a non exhaustive list of the many proposed approaches.

Despite the variety of methods proposed for infering DPPs , the basic properties of the maximum likelihood estimator have received little attention. Exceptions not directly related to our work are Lavancier et al. (2015); Biscio & Lavancier (2017), which make strong parametric assumptions, and Baraud (2013) which makes smoothness assumptions. More closely, the authors in Brunel et al. (2017) study the properties of the expected log-likelihood function and show that it has exponentially many saddle points. They establish a condition under which the global optima have a strictly positive, but arbitrarily small, strong convexity

constant. Finally, they prove several convergence rates for the maximum likelihood estimator. Our approach differs in that we consider a finite number of samples without making any additional specific assumptions. In other words, we work with the finite-sample likelihood rather than the expected likelihood function. Let us also mention that other approaches exist such as in Urschel et al. (2017); Brunel & Urschel (2024), which are based on moments and the principal minor assignment problem.

## 1.2 Organization of the paper

The addressed problem is described in Section 2, with introductory elements about DPPs. Novel properties about the maximum likelihood estimator in the case of a finite sample set are stated in Section 3. The proposed regularization and corresponding optimization algorithms are then described in Sections 4 and 5. Simulation results are given in Section 6, and Section 7 concludes the paper.

Notation:

- Throughout all the paper, $N > 1$ is a fixed integer and we define the set $\mathcal{X} = \{1, \ldots, N\}$. For any set $X \subseteq \mathcal{X}$, its cardinality is $|X|$.

- For any subset $X = \{y_1, \ldots, y_{|X|}\} \subseteq \mathcal{X}$, where $y_1 < \cdots < y_{|X|}$, we define a selection matrix $U_X \in \mathbb{R}^{N \times |X|}$ by $U_X = [e_{y_1}, \ldots, e_{y_{|X|}}]$ where $e_k$ is the $k^{\text{th}}$ unit vector of $\mathbb{R}^N$ (i.e., element of its canonical basis).

  For any matrix $M \in \mathbb{R}^{N \times N}$, and any subsets $R, C \subseteq \mathcal{X}$, we denote $M_{R,C} \in \mathbb{R}^{|R| \times |C|}$ the submatrix obtained by keeping in $M$ rows indexed in $R$ and columns indexed in $C$. We will use the shorthands $M_{R,:}$ when all columns are kept ($C = \mathcal{X}$), $M_{:,C}$ when all rows are kept ($R = \mathcal{X}$), and finally $M_X = M_{X,X}$ for any principal submatrix with columns and rows given by $X \subseteq \mathcal{X}$. Accordingly, we have $M_X = U_X^\top M U_X$ and $M_{X,:} = U_X^\top M$.

- $\mathbb{S}^N$ denotes the space of real-valued symmetric matrices of size $N \times N$, while $\mathbb{S}_{++}^N$ is the cone of positive definite matrices in $\mathbb{S}^N$. For any matrix $M$, $\det(M)$ is its determinant. If $M \in \mathbb{S}^N$, $\lambda_k^M$ is its $k^{\text{th}}$ real eigenvalue, ordered by increasing values. Finally, $\mathbf{I}_N$ is the identity matrix of size $N \times N$.

## 2 Problem statement

### 2.1 Determinantal point processes

In this section, we introduce some definitions and properties about DPPs (see Kulesza & Taskar (2012) for a more detailed introduction). We will consider DPPs only on the finite ground set $\mathcal{X}$. A DPP on $\mathcal{X}$ is a random variable $\mathbf{X}$ with values in the set $2^{\mathcal{X}}$ of all subsets of $\mathcal{X}$, such that

$$(\forall A \subseteq \mathcal{X}) \qquad \mathbb{P}(\mathbf{X} \supseteq A) = \det(K_A). \tag{1}$$

The above definition requires a matrix $K \in \mathbb{R}^{N \times N}$ called the marginal kernel of the DPP whose components are indexed by the elements of $\mathcal{X}$. Then, $K_A = [K_{i,j}]_{i,j \in A}$ is the principal submatrix of $K$ with indices in the set $A$. For simplicity, we assume in this paper that $K$ is symmetric as well as satisfying a necessary condition for its validity: $\mathbf{0} \preceq K \preceq \mathbf{I}_N$.

From this definition, one can see that the diagonal elements of $K$ are $\mathbb{P}(i \in \mathbf{X}) = K_{i,i}$, the marginal probabilities of inclusion in $\mathbf{X}$ for individual elements of $\mathcal{X}$. In addition,

$$\mathbb{P}(\mathbf{X} \supseteq \{i, j\}) = \begin{vmatrix} K_{i,i} & K_{i,j} \\ K_{j,i} & K_{j,j} \end{vmatrix} = \mathbb{P}(i \in \mathbf{X})\mathbb{P}(j \in \mathbf{X}) - K_{i,j}^2 .$$

This proves that the probability of co-occurrence of two elements is smaller than in the independent case, especially for large values of $K_{i,j}$, which corresponds to highly similar items $i$ and $j$. DPPs are hence diversifying in the sense that they tend not to select similar items.

Manipulating inclusion probabilities as in (1) is more complicated than manipulating the joint probability distribution. Whenever all eigenvalues of $K$ are strictly smaller than 1, another slightly less general definition can be given. For a fixed symmetric positive semi-definite matrix $L \in \mathbb{R}^{N \times N}$, we call $L$-ensemble a random variable $\mathbf{X}$ with values in $2^{\mathcal{X}}$ such that

$$(\forall A \subseteq \mathcal{X}) \qquad \mathbb{P}(\mathbf{X} = A) = \frac{\det(L_A)}{\det(L + \mathbf{I}_N)}, \tag{2}$$

where the denominator is the normalizing factor to ensure unit mass of the probability distribution. It is known that an $L$-ensemble is a DPP and its marginal kernel is

$$K = L(L + \mathbf{I}_N)^{-1} = I - (L + \mathbf{I}_N)^{-1}. \tag{3}$$

A DPP with marginal kernel $K$ is an $L$-ensemble only if $\mathbf{I}_N - K$ is invertible. In this case, $L = K(\mathbf{I}_N - K)^{-1}$. A thorough detailed description of the set of all DPPs has been discussed in Tremblay et al. (2023): another representation corresponding to the set of all DPPs has been introduced and called extended $L$-ensembles. In particular, it has been proved that the limits of some $L$-ensembles are no more $L$-ensembles, but extended $L$-ensembles or DPPs. More precisely, extended $L$-ensembles can represent DPPs which have a strictly positive number of elements with probability one (or equivalently, the probability of the emptyset is zero). This is in contrast with $L$-ensembles for which the emptyset having zero probability is only a limit case. In this paper, we tackle the problem of parameter estimation based on the $L$-ensemble description, which is more natural for expressing probabilities and formulating the maximum likelihood problem. However, the mentionned fact about cardinality of $L$-ensembles guided us in this work, as it will appear in the next section.

## 2.2 Maximum likelihood (ML) criterion and estimation

Our learning task is to fit a DPP kernel $L$ to a collection of observed subsets. Let our training data be $X_1, \ldots, X_n \subseteq \mathcal{X}$ which are supposed to be independent and identically distributed and to follow a DPP distribution. We want to estimate a DPP kernel $L$ consistent with this dataset. For all $X \subseteq \mathcal{X}$, let us define $p_X = \mathbb{P}(\mathbf{X} = X)$ and its empirical counterpart $\hat{p}_X = \frac{1}{n} \sum_{i=1}^{n} \mathbb{1}(X_i = X)$ which is the empirical probability of occurrence of each set $X \subseteq \mathcal{X}$ in the training samples. The likelihood is $\prod_{i=1}^{n} \frac{\det(L_{X_i})}{\det(L + \mathbf{I}_N)}$ and the normalized negative log-likelihood hence reads:

$$f_{\mathrm{ML}}(L) = \sum_{X \subseteq \mathcal{X}} \hat{p}_X \left[ \log \det(L + \mathbf{I}_N) - \log \det(L_X) \right]. \tag{4}$$

Note that the above criterion is lower-bounded by zero since it is the negative logarithm of a probability between 0 and 1. Whenever it exists, the maximum likelihood estimator is given by any minimizer of $f_{\mathrm{ML}}$. As a difference of convex (DC) functions, $f_{\mathrm{ML}}$ is not convex and the corresponding optimization problem was recently proved to be $\mathcal{NP}$-hard in Grigorescu et al. (2022), as conjectured in Kulesza (2012). Let us first investigate properties of the maximum likelihood estimator, before considering optimization algorithmic issues.

# 3 Properties of ML estimator

The matrix defining an $L$-ensemble is required to be positive semi-definite. However, when it comes to minimizing $f_{\mathrm{ML}}$ in (4), the stronger condition that $L$ is positive definite is often assumed. Also, the existence of a well defined minimizer (i.e., the existence of a maximum likelihood estimator) is often overlooked. We first prove some properties which apply to any finite set of samples, which will provide insight on this question. It will then appear that, in applications, an additional penalization term can be beneficial.

## 3.1 Non coercive behavior

In order to define a maximum likelihood estimator, the minimum value of the criterion $f_{\mathrm{ML}}$ should be reached. A sufficient condition for this is that the criterion is coercive[1]. We show that unfortunately neither of these

---

[1] This means that $f_{\mathrm{ML}}(L) \to +\infty$ when $\|L\|_{\mathrm{F}} \to +\infty$.

conditions are guaranteed to be satisfied, based on the specific analysis of the case when two elements are in the groundset $\mathcal{X}$.

**Proposition 1** *Suppose that the groundset $\mathcal{X}$ contains $N = 2$ elements. If $\hat{p}_\emptyset = 0$, that is if $X_i \neq \emptyset$ for all $i \in \{1, \ldots, n\}$, then, for any given positive definite matrix $L$, there exists $\tilde{L}$ such that*

$$f_{\mathrm{ML}}(\tilde{L}) < f_{\mathrm{ML}}(L) \quad and \quad \|\tilde{L}\|_{\mathrm{F}} > \|L\|_{\mathrm{F}}.$$

*This implies that the neg-loglikelihood (4) is non coercive.*

The proof in Appendix A relies on constructing the matrix $\tilde{L}$ such that all principal minors of $L$ are scaled by a factor $\alpha > 1$. As a consequence, the criterion $f_{\mathrm{ML}}$ does not reach its minimum and no maximum likelihood estimator exists whenever $\hat{p}_\emptyset = 0$. Intuitively, this is in accordance with the fact that, for $L$-ensembles, the empty set has necessarily strictly positive probability, corresponding to the usual convention $\det(L_\emptyset) = 1$. This is in contrast with DPPs, which are a slightly more general model allowing zero probability for the emptyset. In order to properly define a maximum likelihood estimator, the empty set should hence be contained in the training data samples.

## 3.2 Necessary/sufficient conditions for existence of a positive definite minimizer

We now give a necessary condition for the criterion $f_{\mathrm{ML}}$ in (4) to have a positive definite minimizer, in the general case of any size $N$ of the groundset $\mathcal{X}$ and of the matrix $L$. The proof in Appendix B relies on simple algebra on determinants.

**Proposition 2** *A positive definite minimizer of the neg-loglikelihood in (4) can only exist if the following two conditions are satisfied:*

$$(i) \bigcup_{i=1}^{n} X_i = \mathcal{X} \qquad (ii) \bigcap_{i=1}^{n} X_i = \emptyset.$$

*Equivalently, a positive definite minimizer of (4) can exist only if, for all elements $i_0 \in \mathcal{X}$, the following two conditions are satisfied:*

$$(i) \ (\exists X \subseteq \mathcal{X}) \ i_0 \in X \ and \ \hat{p}_X > 0 \qquad (ii) \ (\exists X \subseteq \mathcal{X}) \ i_0 \notin X \ and \ \hat{p}_X > 0.$$

Note that in Kawashima & Hino (2023), it has been observed that condition (i) above is necessary for avoiding convergence issues of the algorithm minimizing $f_{\mathrm{ML}}$. From the above statement, it is now clear that this phenomenon happens for fundamental reasons and not for numerical ones. Below is a sufficient condition for $f_{\mathrm{ML}}$ to be coercive, which is proved in Appendix C.

**Proposition 3** *If $\hat{p}_\emptyset > 0$, then the negative log-likelihood $f_{\mathrm{ML}}$ in (4) is coercive.*

## 3.3 Particular case with explicit minimization

We now consider the particular case when only $n = 1$ sample $X_1$ is available in the training set. In this simple situation, which has been considered in another context in Barthelmé et al. (2019), the minimum value of $f_{\mathrm{ML}}$ can be explicitly characterized. This will help us both understanding the properties of $f_{\mathrm{ML}}$ and its minima and, later, the behavior of optimization algorithms. Note that the conditions of Proposition 2 are not fulfilled and, as expected, we will show that a maximum likelihood estimator does not exist.

The maximum likelihood criterion can be written in terms of the eigendecomposition $L = V\Lambda V^\top$, where $V$ is orthogonal ($VV^\top = V^\top V = \mathbf{I}_N$) and $\Lambda = \mathrm{Diag}\left((\lambda_i^L)_{1 \leq i \leq N}\right)$. Since $\det(V\Lambda V^\top + \mathbf{I}) = \det(\Lambda + \mathbf{I})$, Equation (4) can be reexpressed as

$$f_{\mathrm{ML}}(L) = \log\det(\Lambda + \mathbf{I}_N) - \sum_{X \subseteq \mathcal{X}} \hat{p}_X \log\det(V_{X,:}\Lambda V_{X,:}^\top). \tag{5}$$

For $n = 1$, a unique term with $\hat{p}_{X_1}$ appears above and minimization with respect to $V$ can be performed explicitly (see Appendix D).

**Proposition 4** *If a single sample $X_1$ of cardinality $r = |X_1|$ is available, the negative loglikelihood $f_{\mathrm{ML}}$ is non coercive. The infimum of $f_{\mathrm{ML}}$ is zero and it is approached by matrices $L$ of rank $r$ and such that the principal submatrix $L_{X_1}$ reads $L_{X_1} = \tilde{V} \operatorname{Diag}\big((\lambda_i^L)_{N-r+1 \le i \le N}\big) \tilde{V}^\top$, where $\tilde{V}$ is any orthogonal matrix and all eigenvalues $(\lambda_i^L)_{N-r+1 \le i \le N}$ go to infinity.*

One can note that the infimum can be approached in a non unique way since there are many possible choices for the matrix $\tilde{V}$. However, it is in particular reached for $\tilde{V}$ being identity and hence for diagonal matrices $L = \Lambda$. Intuitively, it seems natural that from a unique sample, one cannot infer about repulsion between different elements and hence one cannot distinguish a diagonal DPP (in fact, a Bernoulli process) from a non diagonal one.

## 4  Regularized criterion and minimization algorithm

A practical consequence of the above results is that the maximum likelihood estimator is likely not to exist in the class of $L$-ensembles when the empty set does not appear in the training data set. A possible solution to circumvent the problem, is to add a term $\mathcal{P}(L)$ called penalization or regularization term. Preferably, the regularization term should promote some prior information on the matrix $L$ to be estimated. Possible choices for $\mathcal{P}(L)$ classically include the $\ell_1$ norm $\|L\|_1 = \sum_{i,j=1}^N |L_{ij}|$, which promotes sparse $L$ or the nuclear norm $\|L\|_{\mathrm{nuc}} = \sum_{i=1}^N |\lambda_i^L|$ which promotes low-rank $L$. In Gartrell et al. (2017; 2018), the regularization is imposed on the columns of a low-rank decomposition and has been justified to avoid large parameter values and overfitting. More precisely, taking a singular value decomposition $L = V\Lambda V^\top$, it reads $\mathcal{P}(L) = \sum_{i=1}^N \frac{1}{\sharp i} \|\sqrt{\lambda_i^L} V_{i,:}\|_2^2 = \sum_{i=1}^N \frac{\lambda_i^L}{\sharp i}$, where $\sharp i = \sum_{X \in \mathcal{X}, i \in X} \hat{p}_X$ is the empirical probability of occurrence of element $i$. This regularization criterion can be seen as a weighted $\ell_1$ norm on the vector of eigenvalues, also sometimes called weighted nuclear norm[2] (see Gu et al. (2016)). Alternatively a regularization term $\log \det(L + \mathbf{I}_N)$ will be considered and interpreted in Section 5. Importantly, the existence of a minimizer is guaranteed if the chosen term $\mathcal{P}(L)$ is coercive, which is the case for the three previous examples. For minimizing the resulting criterion, we propose to use a Forward-Backward algorithm (FB) which is well-known and widely used in proximal optimization, with a broad range of applications in inverse problems and machine learning (Combettes & Pesquet, 2021; Sra et al., 2012). It is suited for minimizing functions which can be split as the sum of two terms, one of them being differentiable and the other convex:

$$f = g + h. \tag{6}$$

It consists in generating a sequence $(L_k)_{k \ge 0}$ by alternating a gradient step and a proximal step as described below in Alg. 1. When the first function is non convex, the following theorem, which is an immediate

---

**Alg. 1** FB algorithm

---
**Initialization:** $L_0 \in \mathbb{S}_{++}^N$
    **for** $k = 0, 1, \ldots$ **do**
        $L_{k+1} = \operatorname{prox}_{\gamma_k h}\left(L_k - \gamma_k \nabla g(L_k)\right)$       with $\gamma_k \in (0, \frac{2}{\alpha})$
    **end for**

---

outcome of Chouzenoux et al. (2014), provides convergence guarantees of FB algorithm. Comments on this theorem are given below.

**Theorem 1** *Suppose that the function to be minimized $f : \mathbb{S}^N \to (-\infty, +\infty]$ satisfies the Kurdyka-Lojasiewicz inequality (KL) and can be split as in (6). Assume also that $g$ is a differentiable function with an $\alpha-$Lipschitzian gradient ($\alpha > 0$) and $h$ is a proper lower semi-continuous convex function. Then, the sequence $(L_k)_{k \ge 0}$ generated by Alg. 1 converges to a critical point of $f$.*

The splitting as in (6) will be detailed next, but note first that there are no convexity assumption and the theorem applies in our case. For convergence guarantees of FB in the nonconvex case, see also Attouch et al.

---

[2]It is incorrectly called a norm, since it is generally non convex because the eigenvalues depend nonlinearly on $L$. Considering such a term is far out of the scope of the paper.

(2011); Attouch & Bolte (2007). The result states convergence to a critical point only and one may expect more powerful guarantees. However, to the best of our knowledge, there exists no result in the literature proving the convergence of proximal methods to a global minimizer when applied to nonconvex criteria. Finally, note that assuming that $f$ safisties (KL) is a very weak requirement which is here satisfied. Indeed, by considering an o-minimal structure which contains $\log, \exp$, and all semi-algebraic functions, one can prove that the regularization functions of interest satisfy (KL) (Bolte et al., 2007).

In Theorem 1, the function $f$ is defined over the Hilbert space $\mathbb{S}^N$. The original maximum likelihood problem however requires minimization over positive semi-definite matrices. As already mentioned, to deal with this constraint, we will consider that the minimization problem is over the set of positive definite matrices. Therefore, we choose to add an indicator function that guarantees to stay in $\mathbb{S}^N_{++}$. Given $\epsilon > 0$ (small and close to zero), we define the closed convex set $\mathcal{S}^N_\epsilon = \{L \in \mathbb{S}^N \mid L \succeq \epsilon \mathbf{I}_N\}$ and its indicator function $\imath_{\mathcal{S}^N_\epsilon} : \mathbb{S}^N \to (-\infty, +\infty]$, given by

$$\imath_{\mathcal{S}^N_\epsilon}(L) = \begin{cases} 0 & \text{if } L \in \mathcal{S}^N_\epsilon \quad (\text{that is, if } L \succeq \epsilon \mathbf{I}_N) \\ +\infty & \text{otherwise.} \end{cases}$$

Finally, given constants $\mu \geq 0, \nu \geq 0$, and $\epsilon > 0$, our algorithm will minimize the function:

$$\begin{aligned} f_{\mu,\nu,\epsilon}(L) &= f_{\mathrm{ML}}(L) + \mu \log \det(L + \mathbf{I}_N) + \nu \mathcal{P}(L) + \imath_{\mathcal{S}^N_\epsilon}(L) \\ &= (1 + \mu) \log \det(L + \mathbf{I}_N) - \sum_{X \subseteq \mathcal{X}} \hat{p}_X \log \det(L_X) + \nu \mathcal{P}(L) + \imath_{\mathcal{S}^N_\epsilon}(L), \end{aligned} \tag{7}$$

Our algorithm applies for any function $\mathcal{P}(L)$ such that its proximity operator is known; more generally $\mathcal{P}(L)$ could be a sum of functions with known proximity operator. Although our algorithm can handle the general case, we will consider for our purpose only one penalization term: either the $\ell_1$ norm ($\nu > 0, \mathcal{P}(L) = \|L\|_1$ and $\mu = 0$), or the nuclear norm ($\nu > 0, \mathcal{P}(L) = \|L\|_{\mathrm{nuc}}$ and $\mu = 0$), or a $\log \det(L + \mathbf{I}_N)$ penalization ($\mu > 0$ and $\nu = 0, \mathcal{P}(L) = 0$). The latter case will be also discussed in Section 5 using another algorithm.

In the following sections, we describe how it is possible to split the function $f_{\mathrm{ML}}$ and the above $f_{\mu,\nu,\epsilon}$, guided by the fact that in Equation (6), the differentiable part may be concave, whereas the part making use of the proximity operator is convex. We successively consider the concave and differentiable part of the criterion $g$ and then the convex part $h$.

### 4.1 Splitting the objective function: concave part and gradient

For splitting the criterion as in (6), we first choose to define the function $g : \mathbb{S}^N \to (-\infty, +\infty]$ as

$$g(L) = \begin{cases} (1 + \mu) \log \det(L + \mathbf{I}_N) & \text{if } L + \mathbf{I}_N \succ \mathbf{0} \\ +\infty & \text{otherwise.} \end{cases} \tag{8}$$

To apply the FB algorithm, the gradient of $g$ is needed. Moreover, its Lipschitz constant $\alpha$ is necessary for choosing step-sizes $(\gamma_k)_{k \in \mathbb{N}}$ that ensure convergence ($\gamma_k \in (0, \frac{2}{\alpha})$). We have the following result, which is proved in Appendix E:

**Proposition 5** *The function $g$ is differentiable in $\mathbb{S}^N_{++}$ with*

$$(\forall L \in \mathbb{S}^N_{++}) \quad \nabla g(L) = (1 + \mu)(L + \mathbf{I}_N)^{-1}.$$

*In addition, the gradient of $g$ is $(1 + \mu)-$Lipschitzian in $\mathbb{S}^N_{++}$.*

### 4.2 Splitting the objective function: convex part and proximity operator

According to the above choice for $g$, the function $g$ in the decomposition of $f_{\mu,\nu,\epsilon} = g + h$ reads

$$h(L) = - \sum_{X \subseteq \mathcal{X}} \hat{p}_X \log \det(L_X) + \nu \mathcal{P}(L) + \imath_{\mathcal{S}^N_\epsilon}(L). \tag{9}$$

In Algorithm 1, evaluation of the proximity operator of $h$ is required for $\gamma > 0$ and $L \in \mathbb{S}^N$, whose definition is

$$\operatorname{prox}_{\gamma h}(L) = \arg \min_{M \in \mathbb{S}^N} \left( h(M) + \frac{1}{2\gamma} \|L - M\|_{\mathrm{F}}^2 \right). \tag{10}$$

Due to the indicator function in $h$, since $\epsilon > 0$, the proximity operator of $h$ necessarily returns a positive definite matrix. This ensures

$$(\forall \gamma > 0) \; (\forall L \in \mathbb{S}^N) \quad \operatorname{prox}_{\gamma h}(L) \succ 0 \,.$$

and the iterates $(L_k)_{k \geq 0}$ defined in Alg. 1 are hence guaranteed to be positive definite. Writing the optimality conditions for the problem in Equation (10) yields an equation seemingly impossible to solve analytically. We propose in the next section to compute the proximity operator numerically by an inner optimization procedure. Note that, fortunately, results in Chouzenoux et al. (2014) are stronger than Theorem 1 and the convergence to a critical point also holds for an inexact FB algorithm, when using an approximation to the proximity operator.

### 4.3 Evaluating prox: dual block-coordinate forward-backward

**Rewriting prox computation** We now propose an algorithm to compute numerically the proximity operator of $h$. Multiplying by $\gamma$, we can rewrite the minimization (10) depending on the chosen regularization. If $\mathcal{P}(L) = \|.\|_1$, it reads:

$$\min_{M \in \mathbb{S}^N} \left( \iota_{\mathcal{S}_\epsilon^N}(M) + \underbrace{\sum_{X \subseteq \mathcal{X}} \gamma \hat{p}_X(-\log\det(U_X^\top M U_X))}_{\sum_{j=1}^J h_j(U_j^\top M U_j)} + \underbrace{\gamma \nu \|M\|_1}_{h_{J+1}(M)} + \frac{1}{2}\|L - M\|_{\mathrm{F}}^2 \right) \tag{11}$$

and if $\mathcal{P}(L) = \|.\|_{\mathrm{nuc}}$), it reads:

$$\min_{M \in \mathbb{S}^N} \left( \left( \gamma \nu \|L\|_{\mathrm{nuc}} + \iota_{\mathcal{S}_\epsilon^N}(L) \right) + \underbrace{\sum_{X \subseteq \mathcal{X}} \gamma \hat{p}_X(-\log\det(U_X^\top L U_X))}_{\sum_{i=j}^J h_j(U_j^\top M U_j)} + \frac{1}{2}\|L - M\|_{\mathrm{F}}^2 \right). \tag{12}$$

In both cases, the criterion takes the same form as in (Abboud et al., 2017, Eq. (25)), where the linear operators inside each function $h_j$ is given by $M \mapsto U_j^\top M U_j$. The Dual Block-Coordinate Forward-Backward (DBCFB) algorithm proposed in this paper leverages this structure and provides an efficient algorithm for minimizing such a function, so computing the proximity operator of a sum of convex functions involving linear operators. The main ingredient required for implementing DBCFB is the proximity operator of each of the above involved functions.

**Proximity operators of the involved terms** The proximity operators of $\iota_{\mathcal{S}_\epsilon^N}$, $-\log\det$, $\|.\|_1$ and $\|.\|_{\mathrm{nuc}}$ are precisely known and recalled in Appendix F. In the second equation above, the sum of the nuclear norm and the indicator has been considered simultaneously as a single term to avoid unnecessary computations, since the prox of this sum is known. Note that additional terms can be added under the condition that the proximity operator is known. This paves the way to considering many additional constraints or penalization that can be added to $h$ or to the initial objective function.

**Description of the algorithm** The DBCFB algorithm is described in Appendix G both in the cases of Equations (11) and (12) (see Sections G.1 and G.2 respectively). Each iteration cycle consists in taking the proximity operator of the first function which involves $\iota_{\mathcal{S}_\epsilon^N}$, taking the proximity operator of one of the functions $h_j$ and performing linear combinations. From the proximity operator involving the indicator functions, which projects onto $\mathcal{S}_\epsilon^N$, the iterates always remain definite positive. In addition, when using the DBCFB algorithm, we systematically used a warm-restart to initialize the variables at the value obtained at the previous outer iteration during prox computation.

### 4.4 Complexity and alternative dual block coordinate forward-backward

The complexity of our algorithm is $O(N^3)$. Indeed, in the previously described DBCFB algorithm, computing the proximity operator of the terms $\imath_{\mathcal{S}_\epsilon^N}$ or $(\gamma\nu\|.\|_{\mathrm{nuc}} + \imath_{\mathcal{S}_\epsilon^N})$ requires diagonalizing the matrix, which costs $O(N^3)$. Our complexity is hence similar to other existing algorithms (Mariet & Sra, 2015; Kawashima & Hino, 2023; Gillenwater et al., 2014) but it remains a challenge for all existing methods. We shortly discuss below potential directions for improving the computational efficiency.

First, the computationally intensive projection can be activated less frequently. Indeed, another possibility consists in incorporating the latter functions in the sum of the functions $h_j$ in Equations (11) (12) (or equivalently in (Abboud et al., 2017, Eq. (25))). This is precisely what has been done for obtaining the algorithm given in (Abboud et al., 2017, Eq. (37)). This allows us to do a smaller number of projections onto $\mathcal{S}_\epsilon^N$ and thus reduce the computational complexity. This algorithm converges to the sought proximity operator, as mentioned in Abboud et al. (2017). Note also that according to the latter work, more sophisticated strategies are allowed in order to activate the most computation demanding proximity operators less frequently, e.g. using quasi-cyclic rules. Other empirical strategies could be developed, such as fine-tuning the inner-loop for approximate convergence. Another option would be to exploit the full possibilities offered by a variable metric in the FB algorithm as in Chouzenoux et al. (2014). To address large $N$ scenarios, a research direction would be to resort to proximal interior point methods (Chouzenoux et al., 2020). Alternatively, stochastic gradient based approaches have been proposed (Osogami et al., 2018). Note finally that for DPPs with very large $N$ but samples with much smaller cardinality, using a low-rank factorization of $L$ could be beneficial at the cost of loosing the global structure of the optimization problem.

## 5 Minimization with $\log\det$ penalization

We now focus on the particular case when $\nu = 0$ in Equation (7), that is the penalization $\mathcal{P}(L)$ is dropped. The FB algorithm described in the previous section remains valid. If we further drop the indicator function $\imath_{\mathcal{S}_\epsilon^N}$, we obtain the criterion

$$f_\mu(L) = (1 + \mu)\log\det(L + \mathbf{I}_N) - \sum_{X \subseteq \mathcal{X}} \hat{p}_X \log\det(L_X)\,, \tag{13}$$

where $\mu > 0$ is a regularization parameter. Up to a scaling of the objective function by $1 + \mu$, adding the $\mu\log\det(L + \mathbf{I}_N)$ regularization term amounts to modifying the empirical probability and replace each $\hat{p}_X$ by $\hat{p}_X/(1 + \mu)$, whereas the probability of $\emptyset$ is set to the value $\mu/(1 + \mu)$ instead of zero. To minimize $f_\mu$, particularly efficient algorithms can be considered as in Mariet & Sra (2015); Kawashima & Hino (2023). The latter algorithms have been designed and used with $\mu = 0$, although according to Section 1, the maximum likelihood estimator is likely not to exist in the class of $L$-ensembles when the empty set does not appear in the training data set. In practice, the iterations are usually stopped when the progress of the objective function falls below a given tolerance, overlooking the difficulty inherent to the objective function.

**Fixed-point algorithm** For minimization of $f_\mu$, we make a few comments about the algorithm in Mariet & Sra (2015), which is referred to as the "fixed-point algorithm", its iterations being given by

$$(\forall k \in \mathbb{N}) \qquad L_{k+1} = L_k - \gamma L_k \nabla f_\mu(L_k) L_k\,. \tag{14}$$

The above iterations were introduced by considering a fixed-point equation after changing the optimization variable and working with $L^{-1}$. Importantly, in doing so, it can be guaranteed that the successive iterates of the algorithm remain positive definite, provided that the initial point $L_0$ is positive definite. The iterations were also interpreted as an MM (Majorization - Minimization) procedure. We further show in Appendix H that the iteration (14) can be interpreted as a variable metric preconditioned gradient descent algorithm. This metric is actually also called affine invariant metric and considering it amounts to endowing the set of positive definite matrices with a Riemannian structure (Boumal, 2023). As an important remark, from Equation (14) and the expression of the gradient in Appendix H, it can be seen that if the initial $L_0$ is diagonal, then all $(L_k)_{k\in\mathbb{N}}$ are diagonal in the sequence provided by the fixed-point algorithm. In this case,

the algorithm converges to a stationary point of the criterion which is a diagonal DPP kernel, that is only a Bernoulli process is identified, which models no repulsion between points, contrary to sought DPP in general.

**Gradient with low-rank factorization** Another algorithm for minimizing $f_\mu$ has been proposed in Gartrell et al. (2017) and consists of gradient iterations combined with a low-rank factorization of the kernel matrix $L = VV^\top$. We have also used this algorithm in our simulations.

## 6  Simulations

We provide some simulations to confirm the properties stated in Section 3, and to illustrate the effectiveness and limitations of the different algorithms for minimizing the considered criteria. More precisely, we considered the fixed-point algorithm in Mariet & Sra (2015), the low-rank algorithm from Gartrell et al. (2017), and our FB algorithm as described in Section 4. We have also tested the algorithm in Kawashima & Hino (2023) and it systematically converged with increased convergence speed to the same result as the fixed-point algorithm. We hence only reported about the latter, since convergence speed of the algorithms is not the main topic addressed in this paper. Keeping in mind the objective of estimating a DPP probability law, we considered the total variation (TV) distance between any two probability distributions $\left(p_X^{(1)}\right)_{X \subseteq \mathcal{X}}, \left(p_X^{(2)}\right)_{X \subseteq \mathcal{X}}$ over subsets of $\mathcal{X}$:

$$d(p_X^{(1)}, p_X^{(2)}) = \frac{1}{2} \sum_{X \subseteq \mathcal{X}} \left| p_X^{(1)} - p_X^{(2)} \right| . \tag{15}$$

In the above definition, $\left(p_X^{(1)}\right)_{X \subseteq \mathcal{X}}, \left(p_X^{(2)}\right)_{X \subseteq \mathcal{X}}$ can also be given by a matrix $L$ according to (2) and it hence also defines a distance to an $L$-ensemble.

**Non coercivity** For $N = 2$, we generated a probability law on the subsets of $\mathcal{X} = \{1, 2\}$ by drawing randomly $\hat{p}_{\{1\}}, \hat{p}_{\{2\}}, \hat{p}_{\{1,2\}} > 0$ that sum up to one and hence $\hat{p}_\emptyset = 0$. To minimize $f_\mu$ in (13), we ran the fixed-point and low-rank algorithms for the values $\mu = 0$, $\mu = 0.01$, and $\mu = 0.05$. The criterion value $f_\mu(L_k)$, the norm of the iterate $\|L_k\|_F$ and the distance to the true law are plotted on Figure 1 as function of the iteration number $k$. The criterion value indeed converges in all cases. However, for $\mu = 0$, the norm of the

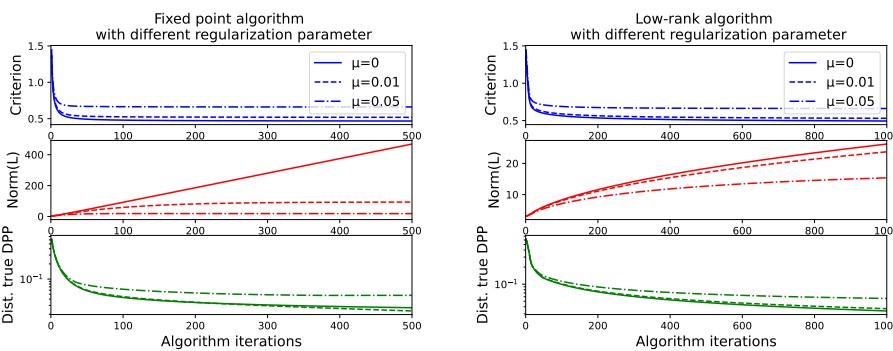

Figure 1: Non-coerciveness of the maximum likelihood criterion for $N = 2$: criterion and distance to true DPP decrease, whereas norm of the iterates increase.

iterates go to infinity, while the distance to the true law to be estimated always decreases. This observation confirms Proposition 1. More generally, we observed experimentally the iterates going to infinity, as soon as $\hat{p}_\emptyset = 0$. In our simulations, the parameter $\mu$ has been manually tuned. To provide a guideline, Figure 2 shows for two randomly drawn DPPs with ground set of cardinality $N = 3$ and $N = 8$ the distance to the true DPP as a function of $\mu$. It seems that an approximately correct value of $\mu$ is $p_\emptyset/(1 - p_\emptyset)$, corresponding to $p_\emptyset = \mu/(1 + \mu)$ as discussed in Section 5.

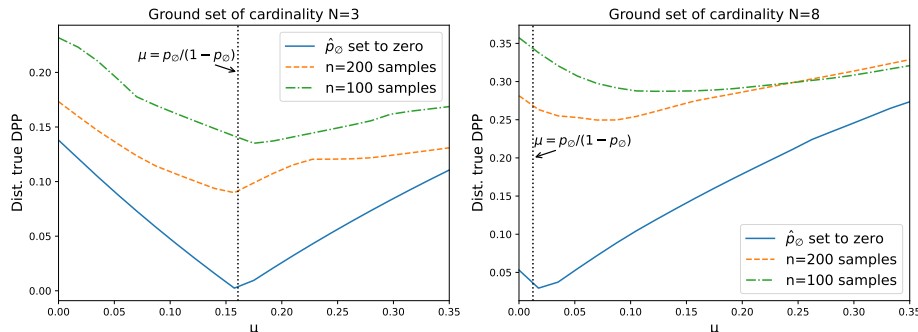

Figure 2: Distance to true DPP as a function of $\mu$ on two examples. The estimated kernel matrix is given by the fixed point algorithm minimization of $f_\mu$. Solid line: artificially set $\hat{p}_\emptyset = 0$ and renormalizing other probabilities. Dashed line: empirical estimation with 200 samples. Dashed-dot line: empirical estimation with 100 samples. Vertical dots indicates $\mu = p_\emptyset/(1 - p_\emptyset)$.

As yet another illustrative example, we reproduced the simulations on the Amazon Baby Registry dataset, as considered in Mariet & Sra (2015) but for many more iterations. We systematically observed that the criterion value converges, whereas the norm of the iterates goes to infinity, as shown on Figure 3 for two categories. The experimental results in this section hence motivate the necessity to add a regularization term in implementations.

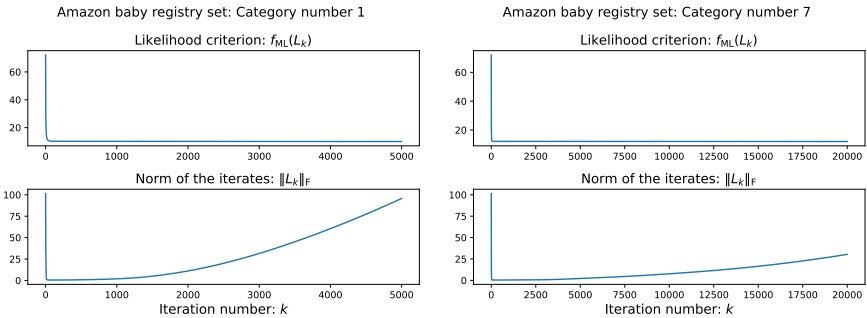

Figure 3: Non-coercivity of the maximum likelihood criterion: the norm of the iterates of the fixed point algorithm on the Amazon Baby Registry dataset diverges, while the criterion value decreases.

**Influence of the indicator function** Our algorithm requires an additional indicator function in $f_{\mu,\nu,\epsilon}$ compared to $f_\mu$. The influence of this indicator and of the choice of $\epsilon$ is illustrated in Table 1 through two examples: denoting $L_\epsilon$ the final value of the algorithm for a given value of $\epsilon$, we reported the relative difference $\|L_\epsilon - L_{10^{-6}}\|_{\mathrm{F}}/\|L_{10^{-6}}\|_{\mathrm{F}}$, where $L_{10^{-6}}$ was considered a reference value. Systematically, we observed that small values of $\epsilon$ only slightly perturb the optimization result, which guided our choice $\epsilon = 10^{-4}$ in the following section.

Table 1: Influence of $\epsilon$ on the final value of the FB algorithm minimizing $f_{\mu,\nu,\epsilon}$: for two randomly driven example ($N = 10$, $n = 50$ samples), the values of $\|L_\epsilon - L_{10^{-6}}\|_{\mathrm{F}}/\|L_{10^{-6}}\|_{\mathrm{F}}$ are reported.

| $\epsilon$ | $1.0 \times 10^{-5}$ | $1.0 \times 10^{-4}$ | $1.0 \times 10^{-3}$ | $1.0 \times 10^{-2}$ | $1.0 \times 10^{-1}$ |
|---|---|---|---|---|---|
| $\mu = 0$ | $2.33 \times 10^{-5}$ | $2.57 \times 10^{-4}$ | $2.60 \times 10^{-3}$ | $2.72 \times 10^{-2}$ | $4.26 \times 10^{-1}$ |
| $\mu = p_\emptyset/(1 - p_\emptyset) \neq 0$ | $2.41 \times 10^{-5}$ | $2.65 \times 10^{-4}$ | $2.71 \times 10^{-3}$ | $2.81 \times 10^{-2}$ | $4.37 \times 10^{-1}$ |

**Behavior of the algorithms with respect to local minima**  For $N = 8$, based on Proposition 4, we illustrate the difficulty of the classical fixed-point algorithm to deal with local minima. Considering a single sample $X_1 \subseteq \mathcal{X}$, we minimized $f_{\mathrm{ML}}$ with the three considered algorithms by initializing the algorithm randomly. We also initialized the algorithms with the identity matrix. The results are shown on Figure 4. We observe that, with random initialization, the fixed-point algorithm systematically converges to a local

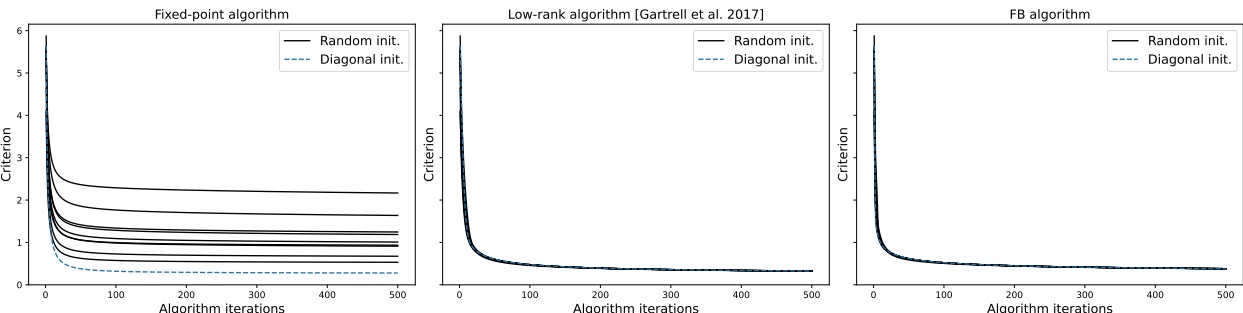

Figure 4: With $n = 1$ sample, for different identical initializations, criterion values of the algorithms depending on iterations. The fixed-point algorithm clearly shows local minima.

minimum and does not approach the global infimum, which is known to be zero. On the contrary, in accordance to the result from Proposition 4 and the remark in the previous subsection, the infimum is approached by the algorithm with a diagonal initialization. In the general case however, such a diagonal initialization should not be used because the algorithm only provides diagonal iterates, which have no reasons to be global minimizers.

We compared the minimization of $f_{\mu,\nu,\epsilon}$ in (7) with $\nu = 0, \epsilon = 10^{-4}$ by our FB algorithm and the minimization of $f_\mu$ in (13) by the fixed-point and low-rank algorithms. Both criteria are hence identical up to the indicator in $f_{\mu,\nu,\epsilon}$. With a random positive definite $L$, we drew $n = 20$ samples according to (2), discarding empty sets. For both algorithms, we have set $\mu = (\det(L + \mathbf{I}_N) - 1)^{-1}$ which amounts to assigning to $\hat{p}_\emptyset$ its theoretical value. We considered the same training set and the same random initialization points for all algorithms (more precisely a Cholesky factor of the same initial $L$ was used for the low-rank algorithm). Figure 5 illustrates our observation that the FB algorithm appears to be less sensitive to the initialization point. We were not able to identify a best initialization strategy for any algorithm, but running the algorithms with a few random initializations appears to slightly improve performance.

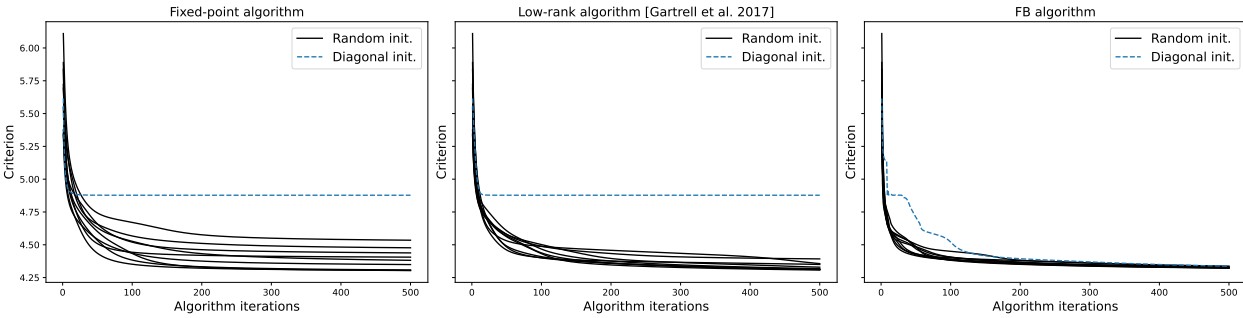

Figure 5: With $n = 20$ sample, for different identical initializations, criterion values of the algorithms depending on iterations. Fixed-point algorithm seems more sensitive to local minima.

**Other penalization**  We drew a random positive definite matrix $L$ of size $N = 20$ and generated $n = 100$ and $n = 300$ DPP samples according to this matrix. From the samples, we tried to infer $L$ and performed minimization of $f_\mu$ in Equation (13) and of $f_{\mu,\nu,\epsilon}$ in Equation (7). In the latter case, we tried both $\ell_1$ and nuclear norm penalization and we choose $\epsilon = 10^{-4}$. We considered the obtained minimizer as an estimate $\hat{L}$

of the true $L$ and quantified the distance between them using the TV distance in (15). The results are given on Figure 6 for $n = 100$ and $n = 300$ samples. One can see that using $\ell_1$ or nuclear norm regularization with an adequate regularization parameter, one is likely to obtain a better estimate.

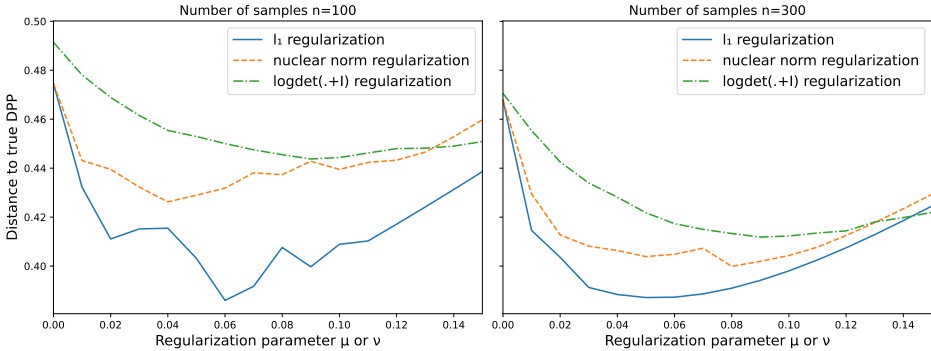

Figure 6: Distance to true DPP depending on regularization choice and parameter $\mu$ or $\nu$.

## 7 Conclusion

In this paper, we addressed the problem of estimating the parameters of a DPP from a set of observed samples. Starting with a maximum likelihood approach, we modeled the samples as realizations of $L$-ensembles. We highlighted significant theoretical challenges, particularly the potential non-existence of the maximum likelihood estimate due to the non-coercivity of the objective function. To overcome this issue, we introduced a regularization term and minimized the resulting criterion using proximal algorithms, which offer flexibility in handling various regularization terms. Our simulations demonstrate the advantages of different penalization strategies and the effectiveness of our approach. Future work should explore ways to accelerate these proximal algorithms through an adaptive choice of the underlying metric.

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

# A    Proof of Proposition 1

Let us write the $2 \times 2$ matrix $L = \begin{bmatrix} x & z \\ z & y \end{bmatrix}$ with $x > 0$, $y > 0$, and $|z| < \sqrt{xy}$. Dropping the term $X = \emptyset$ for which $p_\emptyset = 0$, criterion (4) reads:

$$
\begin{aligned}
f_{\mathrm{ML}}(L) =& \log(1 + x + y + xy - z^2) - \hat{p}_{\{1\}} \log(x) - \hat{p}_{\{2\}} \log(y) - \hat{p}_{\{1,2\}} \log(xy - z^2) \\
=& \hat{p}_{\{1\}} \log(1 + \frac{1}{x} + \frac{y}{x} + \frac{xy - z^2}{x}) + \hat{p}_{\{2\}} \log(1 + \frac{x}{y} + \frac{1}{y} + \frac{xy - z^2}{y}) \\
& + \hat{p}_{\{1,2\}} \log(1 + \frac{x}{xy - z^2} + \frac{y}{xy - z^2} + \frac{1}{xy - z^2}).
\end{aligned}
$$

For $\alpha > 1$, consider $\tilde{L} = \begin{bmatrix} \tilde{x} & \tilde{z} \\ \tilde{z} & \tilde{y} \end{bmatrix}$ where $\tilde{x} = \alpha x$, $\tilde{y} = \alpha y$ and $\tilde{z} = \sqrt{\alpha(\alpha - 1)xy + \alpha z^2}$. We have then $\det(\tilde{L}) = \alpha(xy - z^2) > 0$. A simple calculus yields

$$
\begin{aligned}
f_{\mathrm{ML}}(\tilde{L}) =& \hat{p}_{\{1\}} \log(1 + \frac{1}{\alpha x} + \frac{y}{x} + \frac{xy - z^2}{x}) + \hat{p}_{\{2\}} \log(1 + \frac{x}{y} + \frac{1}{\alpha y} + \frac{xy - z^2}{y}) \\
& + \hat{p}_{\{1,2\}} \log(1 + \frac{x}{xy - z^2} + \frac{y}{xy - z^2} + \frac{1}{\alpha(xy - z^2)}).
\end{aligned}
$$

One can then check that $f_{\mathrm{ML}}(\tilde{L}) < f_{\mathrm{ML}}(L)$ and, in addition, $\|\tilde{L}\|_{\mathrm{F}} \to +\infty$ as $\alpha \to +\infty$. With $\alpha$ large enough, we hence can choose $\tilde{L}$ as stated in Proposition 1.

# B    Proof of Proposition 2

Suppose that there is an element $i_0 \in \mathcal{X} \setminus \cup_{i=1}^n X_i$. Equivalently, $\hat{p}_X = 0$ for all $X \subseteq \mathcal{X}$ that contain $i_0$. One can see that the right term in (4) does not depend on the $i_0$-th row $L_{i_0,:}$ and column $L_{:,i_0}$, that is

$$
f_{\mathrm{ML}}(L) = \log \det(L + \mathbf{I}_N) + \text{terms independent of } L_{i_0,:}/L_{:,i_0}
$$

We use now the linearity of the determinant with respect to the $i_0$-th row and column, so that $\det(L + \mathbf{I})$ is minimum when both the $i_0$-th row and column are zero. With no loss of generality, let us write it for $i_0 = 1$:

$$\det(L + \mathbf{I}) = \det \begin{bmatrix} L_{11} + 1 & L_{12} & \dots & L_{1N} \\ L_{21} & L_{22} + 1 & & \\ \vdots & & \ddots & \\ L_{N1} & \dots & & L_{NN} + 1 \end{bmatrix}$$

$$= \det \begin{bmatrix} L_{11} & L_{12} & \dots & L_{1N} \\ L_{21} & L_{22} + 1 & & \\ \vdots & & \ddots & \\ L_{N1} & \dots & & L_{NN} + 1 \end{bmatrix} + \det \begin{bmatrix} 1 & L_{12} & \dots & L_{1N} \\ 0 & L_{22} + 1 & & \\ \vdots & & \ddots & \\ 0 & \dots & & L_{NN} + 1 \end{bmatrix}$$

$$= \det \begin{bmatrix} L_{11} & L_{12} & \dots & L_{1N} \\ L_{21} & L_{22} + 1 & & \\ \vdots & & \ddots & \\ L_{N1} & \dots & & L_{NN} + 1 \end{bmatrix} + \det \begin{bmatrix} L_{22} + 1 & L_{23} & \dots & L_{2N} \\ L_{32} & L_{33} + 1 & & \\ \vdots & & \ddots & \\ L_{N2} & \dots & & L_{NN} + 1 \end{bmatrix}$$

The above right term does not depend on the first row/column, whereas the determinant on the left is nonnegative and vanishes when the first row/column is zero. Having a zero column and row yields a non definite positive matrix. Condition (i) in the above proposition is hence necessary for a positive definite minimizer.

Condition (ii) is obtained by considering the complement $L$-ensemble, the kernel of which is given by $L^{-1}$.

## C   Proof of Proposition 3

Remind that by convention, the determinant of an empty matrix $\det(L_\emptyset)$ is one (see e.g., Kulesza & Taskar (2012)). Now, all the terms in the sum of Equation (4) are nonnegative since the brackets are the negative logarithm of a probability. Therefore, keeping only the term with $X = \emptyset$ yields

$$f_{\mathrm{ML}}(L) \geq \hat{p}_\emptyset \log \det(L + \mathbf{I}_N) = \hat{p}_\emptyset \sum_{k=1}^{N} \log(1 + \lambda_k^L).$$

Under the assumption $\hat{p}_\emptyset > 0$, the term on the right hand-side goes to infinity as $\|L\| \to \infty$.

## D   Proof of Proposition 4

Remind that $f_{\mathrm{ML}}(L) = \log \det(\Lambda + \mathbf{I}) - \log \det(L_{X_1}) = \log \det(\Lambda + \mathbf{I}) - \log \det\left(V_{X_1,:} \Lambda V_{X_1,:}^\top\right)$. Using the Cauchy-Binet formula, we obtain:

$$\det(V_{X_1,:} \Lambda V_{X_1,:}^\top) = \sum_{S \subseteq \mathcal{X}, |S|=r} \det(V_{X_1,S})^2 \det(\Lambda_{S,S}) \leq \underbrace{\sum_{S \subseteq \mathcal{X}, |S|=r} \det(V_{X_1,S})^2}_{\det(V_{X_1,:} V_{X_1,:}^\top)=1} \left( \prod_{i=N-r+1}^{N} \lambda_i^L \right)$$

$$\leq \prod_{i=N-r+1}^{N} \lambda_i^L.$$

In addition, equality in the above upper-bound is obtained by choosing the matrix $V_{X_1,:}$ block-wise as $V_{X_1,:} = \begin{bmatrix} \mathbf{0} & \tilde{V} \end{bmatrix}$, where $\mathbf{0} \in \mathbb{R}^{r \times N-r}$ has only zero entries and $\tilde{V} \in \mathbb{R}^{r \times r}$ is orthogonal, i.e., $\tilde{V}\tilde{V}^\top = \mathbf{I}_r$. As a consequence, taking $V_{X_1,:}$ as defined above, the following lower-bound for $f_{\mathrm{ML}}$ is obtained:

$$f_{\mathrm{ML}}(L) \geq \sum_{i=1}^{N} \log(1 + \lambda_i^L) - \sum_{i=N-r+1}^{N} \log(\lambda_i^L) = \sum_{i=1}^{N-r} \log(1 + \lambda_i^L) + \sum_{i=N-r+1}^{N} \log(1 + \frac{1}{\lambda_i^L}).$$

The above lower bound is reached for a matrix $L$ of the form given in the proposition and it can be seen that the infimum of the lower bound can only be approached with $\lambda_i^L = 0$ for $i \in \{1, \ldots, N - r\}$ and $\lambda_i^L \to +\infty$ for $i \in \{N - r + 1, \ldots, N\}$. This concludes the proof.

## E    Proof of Proposition 5

The derivation of the gradient follows from classical calculations. For convenience, let us consider the function $\frac{1}{1+\mu}g$. Denoting by $\|.\|_F$ the Frobenius norm and by $\|.\|_{op}$ the spectral norm, we have for $L, R \in \mathbb{S}_{++}^N$:

$$\frac{1}{1+\mu}\|\nabla g(L) - \nabla g(R)\|_F = \|(L + \mathbf{I}_N)^{-1} - (R + \mathbf{I}_N)^{-1}\|_F$$
$$= \|(L + \mathbf{I}_N)^{-1}[(R + \mathbf{I}_N) - (L + \mathbf{I}_N)](R + \mathbf{I}_N)^{-1}\|_F$$
$$= \|(L + \mathbf{I}_N)^{-1}(L - R)(R + \mathbf{I}_N)^{-1}\|_F$$
$$\leq \|(L + \mathbf{I}_N)^{-1}\|_{op} \|L - R\|_F \|(R + \mathbf{I}_N)^{-1}\|_{op}$$
$$\leq \|\mathbf{I}_N\|_{op} \|L - R\|_F \|\mathbf{I}_N\|_{op}$$
$$= \|L - R\|_F.$$

This completes the proof.

## F    Proximity operators

We recall some known proximity operators. A useful reference is for example Bauschke & Combettes (2017). Let $q$ be a positive integer and consider real-valued symmetric matrices of size $q$. Define the operator forming a diagonal matrix $\text{Diag}(.)$ and, for any real-valued symmetric matrix $A \in \mathbb{S}^q$, write its diagonalized form

$$A = P\left[\text{Diag}(\lambda_k)_{k=1}^q\right]P^\top, \tag{16}$$

where the $(\lambda_k)_{k=1}^q$ are its eigenvalues and $P \in \mathbb{R}^{q \times q}$ is orthogonal.

**Property 1** Let $h : \mathbb{S}^q \to (-\infty, +\infty]$ given by

$$h(M) = \begin{cases} -\log\det(M) & \text{if } M \succ 0 \\ +\infty & \text{otherwise} \end{cases}$$

Write $A \in \mathbb{S}^q$ as in Equation (16). Then, for any $\gamma > 0$,

$$\text{prox}_{\gamma h}(A) = \frac{1}{2}P\left[\text{Diag}\left(\lambda_k + \sqrt{\lambda_k^2 + 4\gamma}\right)_{k=1}^q\right]P^\top.$$

**Property 2** Let $A \in \mathbb{S}^q$ and write it as in Equation (16). Then,

$$\text{prox}_{\iota_{\mathcal{S}_\epsilon^N}}(A) = P\left[\text{Diag}\left(\max\{\lambda_k(A), \epsilon\}\right)_{1 \leq k \leq q}\right]P^\top.$$

Note that for any $\gamma > 0$, scaling the indicator function by $\gamma > 0$ has no consequence $(\iota_{\mathcal{S}_\epsilon^N} = \gamma\iota_{\mathcal{S}_\epsilon^N})$ and hence $\text{prox}_{\iota_{\mathcal{S}_\epsilon^N}}(A) = \text{prox}_{\gamma\iota_{\mathcal{S}_\epsilon^N}}(A)$.

**Property 3** Let $A \in \mathbb{S}^q$ and write it as in Equation (16). Then, the proximity operator of the function $\mathbb{S}^q \to (-\infty, +\infty]$ defined by $L \mapsto \nu\|L\|_{\text{nuc}} + \iota_{\mathcal{S}_\epsilon^N}(L)$ (where $\nu > 0$) is given by:

$$\text{prox}_{(\nu\|.\|_{\text{nuc}}+\iota_{\mathcal{S}_\epsilon^N})}(A) = P\left[\text{Diag}\left(\max\{\lambda_k(A) - \nu, \epsilon\}\right)_{1 \leq k \leq q}\right]P^\top$$

# G   Details on the DBCFB algorithms

We give more details on the DBCFB algorihms allowing us to compute the proximity operator in (10). In each case, it is guaranteed that the corresponding sequence $(\Xi_k)_{k\in\mathbb{N}}$ converges to $\operatorname{prox}_{\gamma h}(L)$. In the following subsections, we also denote $N_j$ the number of columns of each matrix $(U_j)_{j=1}^J$ that appears in (11) or (12).

## G.1   DBCFB, case of Equation (11)

The algorithm described here corresponds to (Abboud et al., 2017, Eq. (34a)-(34b)), when applied for the minimization problem (11). The function $\imath_{\mathcal{S}_\epsilon^N}$ in this paper corresponds to the function $f$ in (Abboud et al., 2017, Eq. (25)) and there are $J+1$ terms involving the functions $h_j$. We choose as a particular case to sweep and activate sequentially the proximity operators of the functions $(h_j)_{j=1}^{J+1}$. The algorithm is given as Algorithm 2 below with the notation of this paper. For consistency, we also have $N_{J+1} = N$ and $U_{J+1} = \mathbf{I}_N$.

---

**Alg. 2** Dual block-coordinate FB, version 1

---

**Initialization:** $\left(Y_0^j\right)_{1\leq j\leq J+1} \in \mathbb{S}^{\sum_{j=1}^{J+1} N_j}$ and $Z_0 = -\sum_{j=1}^{J+1} U_j Y_0^j U_j^\top$

    **for** $k = 0, 1, \dots$ **do**

        $\Xi_k = \operatorname{prox}_{\imath_{\mathcal{S}_\epsilon^N}}(L + Z_k)$

        $j_k = k \mod (J+1)$

        $c_k \in (\epsilon, 2-\epsilon)$

        $\tilde{Y}^{j_k} = Y_k^{j_k} + c_k\, U_{j_k}^\top \Xi_k U_{j_k}$

        $Y_{k+1}^{j_k} = \tilde{Y}^{j_k} - c_k\, \operatorname{prox}_{\frac{\gamma}{n c_k} h_{j_k}}\left(\frac{1}{c_k}\tilde{Y}^{j_k}\right)$

        $Y_{k+1}^j = Y_k^j \quad (\forall j \neq j_k)$

        $Z_{k+1} = Z_k - U_{j_k}\left(Y_{k+1}^{j_k} - Y_k^{j_k}\right) U_{j_k}^\top$

    **end for**

---

## G.2   DBCFB, case of Equation (12)

The algorithm described here corresponds to (Abboud et al., 2017, Eq. (34a)-(34b)), when applied for the minimization problem (12). The function $\gamma\nu\|.\|_{\mathrm{nuc}} + \imath_{\mathcal{S}_\epsilon^N}$ in this paper corresponds to the function $f$ in (Abboud et al., 2017, Eq. (25)) and there are $J$ terms involving the functions $h_j$. We choose as a particular case to sweep and activate sequentially the proximity operators of the functions $(h_j)_{j=1}^{J+1}$. The procedure is given in Algorithm 3 below with the notation of this paper.

---

**Alg. 3** Dual block-coordinate FB, version 1

---

**Initialization:** $\left(Y_0^j\right)_{1\leq j\leq J} \in \mathbb{S}^{\sum_{j=1}^{J} N_j}$ and $Z_0 = -\sum_{j=1}^{J} U_j Y_0^j U_j^\top$

    **for** $k = 0, 1, \dots$ **do**

        $\Xi_k = \operatorname{prox}_{\gamma\nu\|.\|_{\mathrm{nuc}} + \imath_{\mathcal{S}_\epsilon^N}}(L + Z_k)$

        $j_k = k \mod J$

        $c_k \in (\epsilon, 2-\epsilon)$

        $\tilde{Y}^{j_k} = Y_k^{j_k} + c_k\, U_{j_k}^\top \Xi_k U_{j_k}$

        $Y_{k+1}^{j_k} = \tilde{Y}^{j_k} - c_k\, \operatorname{prox}_{\frac{\gamma}{n c_k} h_{j_k}}\left(\frac{1}{c_k}\tilde{Y}^{j_k}\right)$

        $Y_{k+1}^j = Y_k^j \quad (\forall j \neq j_k)$

        $Z_{k+1} = Z_k - U_{j_k}\left(Y_{k+1}^{j_k} - Y_k^{j_k}\right) U_{j_k}^\top$

    **end for**

---

# H    Comments on the fixed-point algorithm of Mariet & Sra (2015)

The fixed point algorithm in Mariet & Sra (2015) can be used for minimizing

$$f_\mu(L) = (1 + \mu) \log \det(L + \mathbf{I}_N) - \sum_{X \in \mathcal{X}} \hat{p}_X \log \det L_X \, .$$

In the original paper, the additional penalization term $\mu \log \det(L + \mathbf{I}_N)$ did not appear, but the adaptation is immediate as shown below. Decomposing $L_X$ as $U_X^\top L U_X$, we can also write

$$f_\mu(L) = (1 + \mu) \log \det(L + \mathbf{I}_N) - \sum_{X \in \mathcal{X}} \hat{p}_X \log \det(U_X^\top L U_X)$$

and the gradient reads

$$\nabla f_\mu(L) = (1 + \mu)(L + \mathbf{I}_N)^{-1} - \sum_{X \in \mathcal{X}} \hat{p}_X U_X (U_X^\top L U_X)^{-1} U_X^\top \, .$$

Now, given a step-size $\gamma > 0$ the iterations of the fixed-point algorithm are given by

$$(\forall k \in \mathbb{N}) \qquad L_{k+1} = L_k - \gamma L_k \nabla f_\mu(L_k) L_k \, , \tag{17}$$

According to Mariet & Sra (2015), when $\mu = 0$, it is guaranteed that the above iteration (17) yields a descent algorithm when $\gamma = 1$. Note that by scaling the objective function and considering $\frac{1}{1+\mu} f_\mu$ instead, one can obtain that the algorithm is a descent algorithm when $\gamma = \frac{1}{1+\mu}$.

The above iterations in (17) can be interpreted as a variable metric preconditioned gradient descent algorithm. If $A$ is a symmetric positive definite matrix and we define the linear operator $\mathcal{A}(N) = A^{-1} N A^{-1}$, let

$$\langle M, N \rangle_\mathcal{A} = \langle M, \mathcal{A}(N) \rangle_\mathrm{F} = \langle M, A^{-1} N A^{-1} \rangle_\mathrm{F} = \mathrm{Tr}(M A^{-1} N A^{-1}). \tag{18}$$

This defines an inner product (see Lemma 1 next in Section H.1) and hence induces a new metric. Let us denote by $\nabla_\mathcal{A} f$ the gradient of $f$ with respect to the metric $\langle ., . \rangle_\mathcal{A}$. We easily deduce from the gradient definitions that $\langle \nabla_\mathcal{A} f(L), . \rangle_\mathcal{A} = \langle \nabla f(L), . \rangle$ and hence we have the relation:

$$\nabla_\mathcal{A} f(L) = A \nabla f(L) A \, .$$

This leads to the corresponding preconditioned gradient descent iterations:

$$(\forall k \in \mathbb{N}) \qquad L_{k+1} = L_k - \gamma \nabla_\mathcal{A} f(L_k) = L_k - \gamma A \nabla f(L_k) A.$$

The iterations in (14) or (17) hence correspond to a variable metric preconditioned gradient descent algorithm where, at each step, the metric is changed by substituting $A$ for $L_k$.

## H.1    Justification of the inner product $\langle ., . \rangle_\mathcal{A}$

**Lemma 1** *For $A \succ \mathbf{0}$, let $\mathcal{A}(N) = A^{-1} N A^{-1}$ and consider $\langle ., . \rangle_\mathcal{A}$ as in (18). Then, $\langle ., . \rangle_\mathcal{A}$ is an inner product.*

Proof: Clearly, $\langle ., . \rangle_\mathcal{A}$ is bilinear and symmetric because $\langle M, N \rangle_\mathcal{A} = \mathrm{Tr}(M A^{-1} N A^{-1}) = \mathrm{Tr}(N A^{-1} M A^{-1}) = \langle N, M \rangle_\mathcal{A}$. In addition, we have:

$$\begin{aligned} \langle M, M \rangle_\mathcal{A} &= \mathrm{Tr}(M(A^{-1} M A^{-1})) \\ &= \mathrm{vec}(M)^\top \mathrm{vec}(A^{-1} M A^{-1}) \\ &= \mathrm{vec}(M)^\top (A^{-1} \otimes A^{-1}) \mathrm{vec}(M) \geq 0 \end{aligned}$$

Above, $\mathrm{vec}(\cdot)$ is the column vector obtained by concatenating the columns of the matrix and $\otimes$ is the Kronecker matrix product. Finally, since $A^{-1} \otimes A^{-1} \succ \mathbf{0}$ for positive definite $A^{-1}$, we have $\langle M, M \rangle_\mathcal{A} = 0$ if and only if $M$ is the zero matrix. This concludes the proof. ∎

