# OpenReview forum: "Kernel Matrix Estimation of a Determinantal Point Process from a Finite Set of Samples: Properties and Algorithms"
_TMLR — Accepted by TMLR_

### Review · Reviewer_2ckL · 2025-11-26

**Summary Of Contributions:**

The paper's core contributions revolve around addressing the numerical instability of Maximum Likelihood (ML) estimation for Determinantal Point Processes (DPPs), which is designed to select subsets that exhibit high volume, i.e., diversity through its determinant. The model is parameterized by a positive semi-definite matrix, often called the kernel matrix $L$, where high off-diagonal values, i.e., high similarity, result in a lower probability of selecting both corresponding items, enforcing the repulsion/diversity property.

The problem was already shown to be NP-hard by Grigorescu et. al. (2002), but in this paper the authors provide a rigorous theoretical diagnosis for the persistent numerical issues seen in prior work, proving that the standard (unregularized) ML objective function for the $L$-ensemble is non-coercive, i.e., unbounded below, under typical conditions, especially when the empty set is not observed in the data. This non-coercive property is intuitively the reason why algorithms often fail to produce a finite, stable kernel matrix $L$ in this setting.

To solve this problem of existence and stability, the paper establishes that regularization is necessary to ensure the ML criterion has a well-defined optimum. Their main algorithmic contribution is the development of a robust, provably convergent optimization scheme based on proximal Forward-Backward splitting (specifically, a Dual Block-Coordinate Forward-Backward approach). This new algorithm efficiently minimizes the regularized log-likelihood, giving a stable, finite estimate for the DPP kernel matrix.

**Audience:**

Yes

**Audience Explanation:**

Although DPPs are not necessarily a broadly applied tool, the paper seems to handle a meaningful challenge for researchers who work on DPP learning and run into issues of numerical instability of the maximum likelihood estimation for L-ensembles. This work provides a clear theoretical explanation for the issue and introduces a proximal optimization framework that offers a stable alternative. For researchers interested in submodular optimization, kernel methods, and probabilistic modeling, e.g., in tasks like document summarization or recommendation systems, this work seems to offer a useful contribution by addressing a known challenge. The topic is fairly specialized, but the results could be quite relevant for researchers working in this area

**Claims And Evidence:**

Yes

**Claims Explanation:**

The claims presented in the submission appear to be well-substantiated by the evidence provided. The paper successfully argues that the standard maximum likelihood estimator is numerically unstable due to the non-coercive property. Thus, the authors propose a regularization approach and introduce a new algorithm along with convergence guarantees. This approach is then empirically complemented by showing good kernel recovery where previous unregularized methods fail.

**Requested Changes:**

There are no suggestions for critical changes that immediately comes to mind, but to strengthen the paper, perhaps the authors could discuss the computational trade-offs of their proximal method, comparing its per-iteration cost with that of simpler but unstable fixed-point iteration. It would also be helpful to motivate regularization more clearly and highlight why the non-coercive property is a more immediate, practical barrier to DPP learning than the known barriers for NP-hardness. Finally, adding a sensitivity analysis in the experiments to show how the choice of the regularization parameter $\lambda$ affects stability, convergence speed, or final kernel accuracy could further strengthen the empirical results, but again, I don't think it's a critical change.

---

### Review · Reviewer_VE63 · 2025-12-16

**Summary Of Contributions:**

The paper studies the problem of the kernel matrix estimation of a determinantal point process from a finite set of samples. Firstly, the paper gives the properties, including the non-coercive behavior, conditions for the existence of the minimizer, and particular cases with explicit minimization. Secondly, the paper provides a forward-backward algorithm for computing the regularized criterion.

**Audience:**

Yes

**Audience Explanation:**

People who are working on the determinantal processes and recommendation systems may be interested in knowing the findings of this paper.

**Claims And Evidence:**

Yes

**Claims Explanation:**

The papers contain theorems and propositions with accurate proof.

**Requested Changes:**

The following are the concerns/questions/requested changes upon this paper:
1. How to choose the regularization parameter of the regularized criterion?
2. The authors claim that, as stated in the title, the contributions are the properties and algorithm. However, are these contributions leading to a better solution to the DPP. Since the regularized criterion is already studied in the literature and the proximal algorithm is also a standard approach for solving the regularized optimization problem, the author is suggested to clarify how this paper can push forward the frontier.
3. In the abstract, the author has the wording "to address this challenge." What is the challenge?
4. Please add more details or discussions for Theorem 1.
5. How does the algorithm behave when N is very large?

---

### Review · Reviewer_bWjR · 2026-01-07

**Summary Of Contributions:**

The paper studies the problem of maximum likelihood estimation (MLE) for Determinantal Point Processes (DPPs), where the kernel matrix is learned without parametric assumptions. The authors show that the standard maximum likelihood objective may fail to admit a solution when the coercivity condition is not satisfied. To address this issue, the paper argues that regularization is essential rather than optional in order to make the optimization problem well posed. The authors propose a regularized ML framework and develop a proximal forward–backward algorithm to solve the resulting optimization problem. Empirical results further demonstrate improved stability and estimation quality when regularization is employed.

**Additional Comments:**

I am not an expert in the DPP literature, and it is possible that I may have missed some important related work that could affect the assessment of this paper’s contribution. I am open to revising my opinion in light of points raised by other reviewers.

**Audience:**

Yes

**Audience Explanation:**

This paper will be of interest to researchers working on problems that can be addressed using Determinantal Point Processes, such as recommendation systems. The proposed approach clearly improves upon previous methods for learning DPPs, while providing strong theoretical guarantees.

**Broader Impact Concerns:**

No concern.

**Claims And Evidence:**

Yes

**Claims Explanation:**

The paper makes strong theoretical and methodological contributions to the problem of learning Determinantal Point Processes (DPPs):

1. The paper clearly demonstrates failure cases of the maximum likelihood objective for L-ensemble DPPs, in particular the non-coercivity of the objective. It provides both empirical evidence illustrating this non-coercive behavior and theoretical analysis identifying conditions under which coercivity may or may not hold.

2. Motivated by the lack of coercivity, the paper theoretically shows that certain regularization terms guarantee coercivity of the objective. As a result, regularization is shown to be necessary rather than optional for well-posed DPP learning.

3. The authors further propose a proximal forward–backward algorithm to solve the resulting regularized MLE problem and empirically demonstrate improved performance compared to existing methods, including reduced sensitivity to initialization and improved behavior with respect to local minima.

**Requested Changes:**

1. For both the proposed method and existing approaches, computational efficiency remains a primary challenge, as these methods typically have $O(N^3)$ complexity. While this does not weaken the contribution of the paper, it would be valuable if the authors could discuss potential directions for improving the computational efficiency or scalability of the algorithm.

2. In the experimental section, it would be helpful if the authors could include additional experiments or discussion regarding the tuning of hyperparameters $\mu, \epsilon, \nu$, to provide guidance on their practical selection.

---

### Decision · Action_Editor_tsch · 2026-02-24

**Recommendation:** Accept as is

**Additional Comments:**

Reviewers agreed on the fact that the authors provide a strong theoretical  contribution to the problem of estimating Determinantal Point Processes, and motivate clearly the interest of a penalization term.

The main concerns of the reviewers  were related to the computational efficiency of the algorithm,  and also the lack of detailed analysis regarding the hyperparameters.  The computational trade-offs of the proximal method and its behavior for large values of N required also additional comments.

It is true that the computational complexity of the proposed approach is its major drawback. However, as mentioned by the authors, it is similar to other existing algorithms. The authors propose directions for improvements in Section 4. They also provided additional guidance through a supplementary simulation (Figure 2) illustrating the choice of mu, along with related comments.

 The additional simulations provide some further insight into the guidelines for selecting the hyperparameters. Even if the paper was improved by these additional discussions, the authors could improve in the camera ready version of the paper the design of the figures (in terms of size and readability) and offer a more detailed discussion of the influence of N.

**Audience:**

Yes

**Audience Explanation:**

Overall, the contribution, the non-coercivity analysis and the optimization is rigorous and is of interest for a small part of the ML community. The numerical contribution is less convinving but support the theoretical claims which in my opinion makes the paper interesting for some of the TMLR'audience even if it's not a perfect fit for most of TMLR readers.

**Claims And Evidence:**

Yes

**Claims Explanation:**

The claims of the authors are supported by various theoretical results a few numerical illustrations. The contributions, which are mostly theoretical, can be summarized as follows.

This paper addresses the problem of Determinantal point processes (DPP) estimation based on finite number of observed subsets. The authors assume that they have access to independent and identically distributed subsets with a DPP distribution. They aim at estimating a DPP kernel consistent with this dataset.

The optimization problem to obtain the maximum likelihood estimator is known to be complex. The authors first propose  to analyze the loss function and establish in a simple setting  that the negative loglikelihood is non coercive which means that no maximum likelihood estimator exists in this setting. Then, they propose a necessary condition for the existence of a positive definite minimizer.

A practical implication of the analysis of the likelihood function is that the maximum likelihood estimator may fail to exist when the empty set is absent from the training dataset. This motivates the introduction of  a penalization or regularization term.
The authors then propose a splitting approach of the negative loglikelihood : a first differentiable concave part, with explicit Lipschitz constant, and a convex part which requires the evaluation of the prox operator, which allows to introduce an algorithm to compute numerically the prox operator.